# Amortized Bayesian Workflow (Extended Abstract)

**Marvin Schmitt**[*]
University of Stuttgart
Germany

**Chengkun Li**[*]
University of Helsinki
Finland

**Aki Vehtari**
Aalto University
Finland

**Luigi Acerbi**
University of Helsinki
Finland

**Paul-Christian Bürkner**
TU Dortmund University
Germany

**Stefan T. Radev**
Rensselaer Polytechnic Institute
United States

## Abstract

Bayesian inference often faces a trade-off between computational speed and sampling accuracy. We propose an adaptive workflow that integrates rapid amortized inference with gold-standard MCMC techniques to achieve both speed and accuracy when performing inference on many observed datasets. Our approach uses principled diagnostics to guide the choice of inference method for each dataset, moving along the Pareto front from fast amortized sampling to slower but guaranteed-accurate MCMC when necessary. By reusing computations across steps, our workflow creates synergies between amortized and MCMC-based inference. We demonstrate the effectiveness of this integrated approach on a generalized extreme value task with 1000 observed data sets, showing efficiency gains (90x faster inference) while maintaining high posterior quality.

## 1 Introduction

In statistics, we often reason about unknown parameters $\theta$ from observables $y$ modeled as a joint distribution $p(\theta, y)$. The posterior $p(\theta \mid y)$ is the statistically optimal solution to this inverse problem, and there are different computational approaches to approximate this costly distribution.

Markov chain Monte Carlo (MCMC) methods constitute the most popular family of sampling algorithms due to their theoretical guarantees and powerful diagnostics [6, 7]. MCMC methods yield autocorrelated draws conditional on a fixed data set $y_{\text{obs}}$. As a consequence, the probabilistic model has to be re-fit for each new data set, which necessitates repeating the entire MCMC procedure from scratch. For such algorithms performed conditionally on a fixed data set, the well-established *Bayesian workflow* [7] defines an iterative sequence of steps that encompasses model specification, fitting, evaluation, addressing computational issues, modifications, and model comparison.

Differently, *amortized Bayesian inference* uses deep neural networks to learn a direct mapping from observables $y$ to the corresponding posterior $p(\theta \mid y)$. Amortized inference follows a two-stage approach: (i) a training stage, where neural networks learn to distill information from the probabilistic model based on simulated examples of observations and parameters $(\theta, y) \sim p(\theta) \, p(y \mid \theta)$; and (ii) an inference stage where the neural networks approximate the posterior distribution for an unseen data set $y_{\text{obs}}$ in near-instant time without repeating the training stage. The Bayesian workflow is not directly transferable to amortized inference because the approximation step is learned over the prior predictive space (see Section 2) while only the inference step is conditional on a fixed data set.

---

[*]equal contribution

Workshop on Bayesian Decision-making and Uncertainty, 38th Conference on Neural Information Processing Systems (NeurIPS 2024).

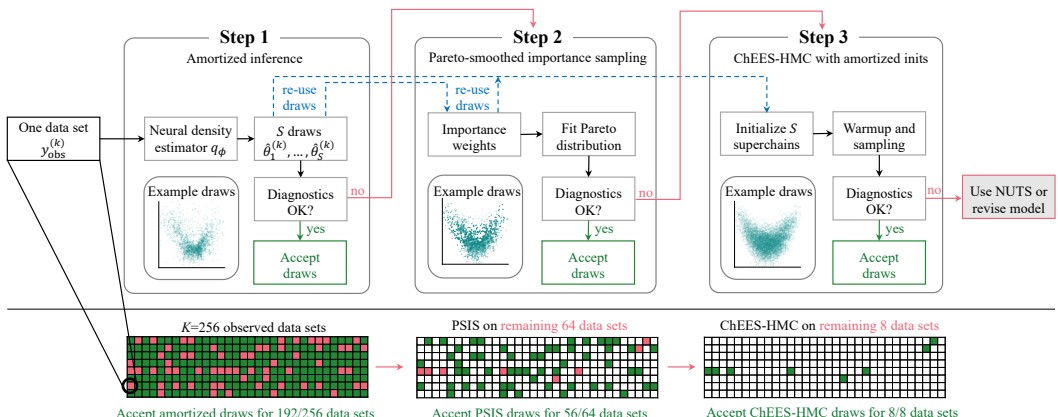

Figure 2: Our adaptive workflow leverages near-instant amortized posterior sampling when possible and gradually resorts to slower – but more accurate – sampling algorithms. As indicated by the blue dashed arrows, we re-use the $S$ draws from the amortized posterior in step 1 for the subsequent steps in the form of PSIS proposals (step 2) and initial values in ChEES-HMC (step 3).

In Bayesian inference, both MCMC (e.g., ChEES-HMC; [9]) and amortized inference lie at the Pareto front of methods that have a favorable trade-off between accuracy and speed. In this paper, we propose an adaptive workflow that yields high-quality posterior draws while minimizing the required compute time by *moving along the Pareto front* to afford fast-and-accurate inference when possible, and slow-but-guaranteed-accurate inference when necessary (see Figure 1). Crucially, our workflow consistently yields high accuracy, as evaluated with tailored diagnostics in all steps. Furthermore, it re-uses computations for subsequent steps in the form of importance sampling proposals and initializations of many-short-chains MCMC. The software implementation encompasses an end-to-end workflow featuring model specification via `PyMC` [15], amortized inference with deep learning via `BayesFlow` [16], and GPU-enabled ChEES-HMC [9] via `Tensorflow Probability` [5].

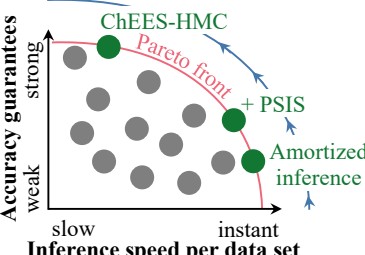

Figure 1: Our workflow adaptively moves along the Pareto front and re-uses previous computations.

## 2 Integrating Amortized Inference into the Bayesian Workflow

Our adaptive workflow starts with neural network training to enable subsequent amortized inference on any number of unseen data sets. While this training phase is conceptually identical to standalone amortized inference training, the inference phase features a principled control flow that guides the analysis based on tailored diagnostics in order to select the appropriate inference algorithm for each observed data set while re-using computations along the way.

### 2.1 Training phase: simulation-based optimization

Since most Bayesian models are generative by design, we can readily simulate $M$ tuples of parameters and corresponding observations from the joint model,

$$(\theta^{(m)}, y^{(m)}) \sim p(\theta, y) \quad \Leftrightarrow \quad \theta^{(m)} \sim p(\theta), \ y^{(m)} \sim p(y \mid \theta) \ \text{for } m = 1, \dots, M \tag{1}$$

which results in the training set $\{(\theta^{(m)}, y^{(m)})\}_{m=1}^M$.[2] The total number $M$ of example tuples is called the *training budget*, and the quality of the amortized posterior estimator hinges on a sufficient training budget. In the case study, we use flow matching [13] as a flexible neural estimators, but our workflow is agnostic to the exact choice of neural network architecture.

**Diagnostics.** At this point, there are no observed data sets yet to guide data-conditional diagnostics. However, we can easily simulate a synthetic *test set* $\{(\theta_*^{(j)}, y^{(j)})\}_{j=1}^J$ of size $J$ from the joint model via Eq. 1. In this *closed-world* setting, we know which "true" parameter vector $\theta_*^{(j)}$ generated each simulated test data set $y^{(j)}$. We evaluate the amortized posterior's bias and variance via

---

[2]This data generation scheme is also known as *prior predictive sampling*.

the normalized root mean-squared error (NRMSE) and perform simulation-based calibration (SBC; [19, 21]) checking to evaluate the uncertainty calibration. These evaluations act as a convergence diagnostic to assert that the neural estimator yields faithful posterior draws under idealized conditions (see Appendix A for details). If these closed-world convergence diagnostics fail, we should tune the training hyperparameters (e.g., training duration, simulation budget, neural network architecture).

## 2.2 Inference phase: posterior approximation on observed data sets

We now use the pre-trained neural network to achieve rapid amortized posterior inference on a total of $K$ observed data sets $\{y_{\text{obs}}^{(k)}\}_{k=1}^K$, which naturally do not come with known ground-truth parameters. The diagnostics in this step are evaluated conditional on each observed data set to determine whether the whole set of amortized draws is acceptable for each specific data set.

### 2.2.1 Step 1: Amortized posterior draws

We aim to use the rapid sampling capabilities of the amortized posterior approximator $q_\phi$ whenever possible according to the diagnostics. Therefore, the natural first step for each observed data set $y_{\text{obs}}^{(k)}$ is to query the amortized posterior and sample $S$ posterior draws $\hat{\theta}_1^{(k)}, \ldots, \hat{\theta}_S^{(k)} \sim q_\phi(\theta \mid y^{(k)})$ in near-instant time (see Figure 2, first panel).

**Diagnostics.** Amortized inference may yield unfaithful results under distribution shifts [11, 20, 23]. Therefore, we assess whether an observed data set is atypical under the data-generating process of the joint model. We define atypical data as data sets that have a larger maximum mean discrepancy (MMD; [8]) to the training set than 95% of the training data sets themselves and frame this decision problem as a sampling-based hypothesis test, as proposed by [20]. The method is illustrated in Figure 3 and formalized in Appendix B. Since the amortized approximator has no accuracy guarantees for data outside of the typical set of the joint

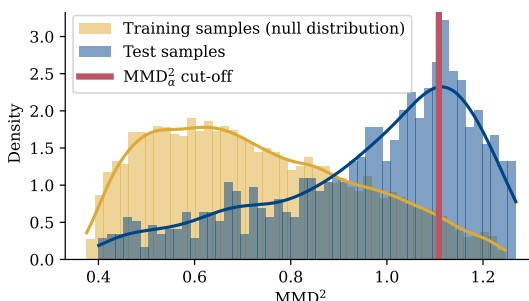

Figure 3: Illustration of our sampling-based hypothesis test that flags atypical data sets where amortized inference has no accuracy guarantees.

model, we propagate such atypical data sets to the next step. Additional data-conditional diagnostics (e.g., posterior predictive checking) can complement our sampling-based atypicality test to evaluate the trustworthiness of the amortized posterior draws.

### 2.2.2 Step 2: Pareto-smoothed importance sampling

As a first pursuit to improve the quality of the amortized posterior draws with a small overhead in computation time, we use a Pareto-smoothed sampling importance sampling (PSIS) scheme [22] (see Figure 2, second panel). Based on the amortized posterior draws from step 1, we compute the importance weights $w_s^{(k)} = p(y^{(k)} \mid \hat{\theta}_s)\, p(\hat{\theta}_s)/q_\phi(\hat{\theta}_s \mid y^{(k)})$ conditional on each observed data set $y^{(k)}$ and smooth the tail of the weight distribution based on fitting a generalized Pareto distribution (aka. Pareto-smoothing; [22]). These smoothed importance weights are then used for computing posterior expectations and for improving the posterior draws with the sampling importance resampling (SIR) scheme [18]. While the utility of standard importance sampling for improving neural posterior draws has previously been investigated [4], we specifically use the PSIS algorithm which is self-diagnosing and therefore better suited for a principled workflow.

**Note.** Common neural architectures for amortized inference (e.g., normalizing flows, flow matching) are mode covering.[3] When the neural network training stage is insufficient (e.g., small simulation budget or poorly optimized network), this may lead to overdispersed posteriors. Fortunately, this errs in the right direction, and PSIS can generally mitigate overdispersed mode-covering draws.

---

[3]Conditional flow matching is mode covering [13]. Normalizing flows are mode covering because they optimize the *forward* KL divergence [17]. In contrast, variational inference algorithms typically optimize the *reverse* KL divergence, which leads to *mode seeking* behavior that is less favorable for importance sampling.

**Diagnostics.** We use the Pareto-$\hat{k}$ diagnostic to gauge the fidelity of the PSIS-refined posterior draws. According to established guidelines [22, 24], Pareto-$\hat{k} \leq 0.7$ indicates good results, whereas $\hat{k} > 0.7$ implies that the draws should be rejected and the respective data sets proceed to step 3.

### 2.2.3 Step 3: ChEES-HMC with amortized initializations

If Pareto-smoothed importance sampling fails according to the diagnostics, we resort to an MCMC sampling scheme which is augmented by re-using computations from the previous steps. Concretely, we use the ChEES-HMC algorithm [9] that affords to launch thousands of parallel chains on a GPU. To accelerate convergence, we use the importance weights from step 2 to sample $S$ (e.g., 16) unique draws for initializing $S$ ChEES-HMC super-chains[4], each with $L$ (e.g., 128) subchains for the nested-$\widehat{R}$ diagnostic below. For the purpose of ChEES-HMC initialization, it is also desirable that the amortized posterior draws are generally mode covering (cf. step 2).

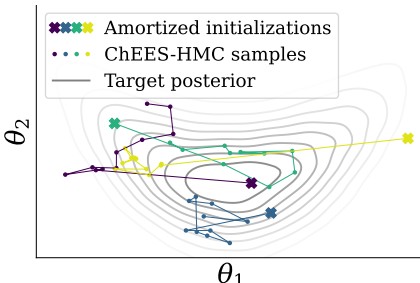

Figure 4: We initialize many ChEES-HMC chains with amortized draws.

**Diagnostics.** In this last step, we use the nested $\widehat{R}$ diagnostic [14] which is specifically designed to assess the convergence of the many-but-short MCMC chains. If the diagnostics in this step indicate unreliable inference, we recommend resorting to the overarching Bayesian workflow [7] and addressing the computational issues that even persist when using the (ChEES-)HMC algorithm. This could involve using the established NUTS-HMC algorithm ([3, 10]) or revising the Bayesian model.

## 3 Empirical Demonstration: Generalized Extreme Value Distribution

In this section, we illustrate the application of the proposed workflow with Bayesian inference on the parameters of a generalized extreme value (GEV) distribution. The GEV distribution is characterized by three parameters: a location parameter $\mu \in \mathbb{R}$, a scale parameter $\sigma \in \mathbb{R}_{>0}$, and a shape parameter $\xi \in \mathbb{R}$, with cumulative distribution function

$$G(y) = \exp\left\{ -\left[ 1 + \xi\left( \frac{y - \mu}{\sigma} \right) \right]^{-1/\xi} \right\}, \tag{2}$$

and we use the prior distributions from Caprani et al. [2] (see Appendix C for details). Given $N = 65$ i.i.d. observations $y = (y_1, \ldots, y_{65})$ from the GEV distribution, we aim to compute a posterior estimate for the data-generating parameters $\theta = (\mu, \sigma, \xi)$. We first train the amortized posterior approximator on simulated parameters and verify that its closed-world performance is satisfactory, as indexed by high parameter recovery and excellent calibration (see Appendix C).

As summarized in Table 1, we perform inference on a total of $K = 1000$ test data sets which are deliberately sampled from a model with a $2\times$ wider prior distribution to emulate out-of-distribution settings in real applications (see Appendix C for details). In step 1, we draw 2000 posterior samples from the amortized approximator $q_\phi$, which takes 150 seconds for all 1000 data sets (2 million posterior draws in total). We confirm that $678/1000$ observed data sets are typical under the data-generating process and accept the amortized draws. The remaining 322 data sets are passed to stage 2, where we apply the PSIS algorithm, taking a total of 130 seconds. The Pareto-$\hat{k}$ diagnostic signals acceptable results for 228 of the 322 data sets, which means that we propagate the remaining 94 data sets to stage 3. Here, we initialize the parallel ChEES-HMC sampler with the amortized draws and observe that the nested $\widehat{R}$ values lie below 1.01 for 66 of the data sets, leading to acceptance of the ChEES draws. This leaves only 28 data sets for separate inference with NUTS-HMC. In total, our amortized Bayesian workflow took $\approx 10$ minutes and led to high-quality posterior draws on all steps, as indicated by a small MMD to a reference posterior. In contrast, running NUTS-HMC on all 1 000 observed test data sets would have taken $\approx 955$ minutes (16 hours), which underscores the efficiency gains of our integrated workflow.

---

[4]If importance sampling resampling without replacement fails to return $S$ valid draws for initializing the chains (e.g., due to less than $S$ non-zero importance weights), we fall back to random initializations.

|  | Accepted datasets | Time | TPA[1] | MMD to reference |
|---|---|---|---|---|
| **Step 1:** Amortized inference | 678/1 000 | 142 | 0.21 | 0.0082 [$4\times10^{-4}$,0.35] |
| **Step 2:** Amortized + PSIS | 228/322 | 124 | 0.54 | 0.0010 [$1\times10^{-4}$,0.02] |
| **Step 3:** ChEES-HMC w/ inits | 66/94 | 398 | 6.03 | 0.0001 [$1\times10^{-5}$,0.05] |
| **Total:** aggregated over steps | 972/1 000 | 664 | 0.68 | — |

[1] TPA: **t**ime **p**er **a**ccepted data set in seconds, computed as the expended time relative to the number of accepted data sets in this step.

Table 1: MMD (median, 95% CI) quantifies the distance between approximate and reference posterior draws. All times are wall-clock seconds on an NVIDIA A100. The time for step 1 includes the training (120s), inference (10s), and diagnostics (12s) stages of the amortized approximator. Our amortized workflow yielded a total of 2 million posterior draws in 11 minutes, whereas using NUTS on all data sets takes approximately 16 hours. While the MMD in step 1 is numerically higher than in steps 2 and 3, spot checks indicated that the posteriors are visually similar to the reference draws.

**Amortized draws can be good ChEES-HMC inits.** To further investigate whether the amortized posterior estimates are indeed beneficial for initializing ChEES-HMC chains, we randomly collect 20 test datasets that are passed to step 3 in the workflow. This indicates that both the amortized posterior draws and their Pareto-smoothed refinement are deemed unacceptable, as quantified by Pareto-$\hat{k} > 0.7$ in step 2. We initialize the ChEES-HMC chains with three different methods: (1) Amortized posterior draws, (2) PSIS-refined amortized draws, and (3) a random initialization scheme similar to Stan [3]. We run the chains for different numbers of warmup iterations followed by a single sampling iteration. As described in Section 2, we use the nested $\widetilde{R}$ value to gauge whether the

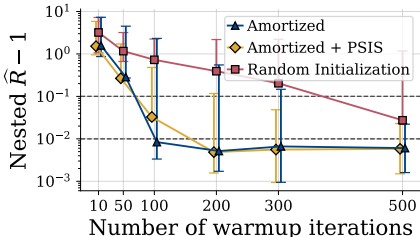

Figure 5: Using amortized posterior draws as inits can reduce the required warmup in ChEES-HMC, but the extent of the benefit varies. The figure shows median$\pm$IQR across 20 test data sets.

chains converged appropriately during the warmup stage (as quantified by common $\widehat{R} - 1$ thresholds of $10^{-1}$ or $10^{-2}$). As shown in Figure 5, amortized posterior draws (and their PSIS-refined counterparts) can significantly reduce the required number of warmup iterations to achieve convergence of ChEES-HMC chains, *even though the draws themselves have previously been flagged as unacceptable*. This emphasizes that our amortized workflow creates synergies by re-using computations in subsequent steps. However, it is not evident whether initializing ChEES-HMC with the PSIS-refined draws from step 2 has an advantage over using the raw amortized draws from step 1, and we mainly see that PSIS improves the worst-case performance (upper error boundary in Figure 5).

## 4 Conclusion

We presented an adaptive Bayesian workflow to combine the rapid speed of amortized inference with the undisputed sampling quality of MCMC in the context of many observed data sets while maintaining a high quality of posterior draws. Our workflow efficiently uses resources by (i) using fast (amortized) inference when the results are accurate; (ii) refining draws with PSIS when possible; and (iii) amortized initializations of slow-but-guaranteed-accurate MCMC chains when needed.

## Acknowledgments

MS and PB acknowledge support of Cyber Valley Project CyVy-RF- 2021-16, the DFG under Germany's Excellence Strategy – EXC-2075 - 390740016 (the Stuttgart Cluster of Excellence SimTech). MS acknowledges travel support from the European Union's Horizon 2020 research and innovation programme under grant agreements No 951847 (ELISE) and No 101070617 (ELSA), and support from the Aalto Science-IT project. CL and LA were supported by the Research Council of Finland (grants number 356498 and 358980 to LA). AV acknowledges the Research Council of Finland Flagship program: Finnish Center for Artificial Intelligence, and Academy of Finland project 340721.

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

# A Closed-world diagnostics

In the following, let $\hat{\theta}_1^{(j)}, \ldots, \hat{\theta}_S^{(j)} \sim q_\phi(\theta \,|\, y^{(j)})$ be $S$ draws from the amortized posterior $q_\phi(\cdot)$.

## A.1 Normalized root mean-squared error

As a measure of posterior bias and variance, we assess the recovery of the ground-truth parameters, for example via the average normalized root mean squared error (RMSE) over the test set,

$$\text{NRMSE} = \frac{1}{J} \sum_{j=1}^{J} \frac{1}{\text{range}(\theta_*)} \sqrt{\frac{1}{S} \sum_{s=1}^{S} \left(\theta_*^{(j)} - \hat{\theta}_s^{(j)}\right)^2}, \tag{3}$$

where $\text{range}(\theta_*) = \max_k(\theta_*^{(k)}) - \min_k(\theta_*^{(k)})$.

## A.2 Simulation-based calibration checking

Simulation-based calibration (SBC; [19, 21]) checking evaluates the uncertainty calibration of the amortized posterior. For the *true* posterior $p(\theta \,|\, y)$, all intervals $U_q(\theta \,|\, y)$ are well-calibrated for any quantile $q \in (0, 1)$ [1],

$$q = \iint \mathbf{I}[\theta_* \in U_q(\boldsymbol{\theta} \,|\, y)] \, p(y \,|\, \theta_*) \, p(\theta_*) \mathrm{d}\theta_* \mathrm{d}y, \tag{4}$$

with indicator function $\mathbf{I}[\cdot]$. Insufficient calibration of the posterior manifests itself as violations of Eq. 4. To quantify these violations, we report the expected calibration error of the amortized posterior, computed as median SBC error of 20 posterior credible intervals with increasing centered quantiles from $0.5\%$ to $99.5\%$, averaged across the $J$ examples in the test set.

# B Testing for atypicality in step 1

Inspired by an out-of-distribution checking method for amortized inference under model misspecification [20], we use a sampling-based hypothesis test to flag atypical data sets where the trustworthiness of amortized inference might be impeded. Concretely, we use the sampling-based estimator for the maximum mean discrepancy (MMD; [8]),

$$\text{MMD}^2(p \,||\, q) = \mathbb{E}_{x,x' \sim p(x)}[\kappa(x, x')] + \mathbb{E}_{x,x' \sim q(x)}[\kappa(x, x')] - 2\mathbb{E}_{x \sim p(x), x' \sim q(x)}[\kappa(x, x')], \tag{5}$$

where $\kappa(\cdot, \cdot)$ is a positive definite kernel and we aim to quantify the distance between the distributions $p, q$ based on samples.

In our case of atypicality detection in step 1, $p$ is the distribution of training data $y$ used during simulation-based training, and $q$ is the opaque distribution behind the observed test data sets. We construct a hypothesis test, where the null hypothesis states that $p = q$. For $M$ training data sets $\{y^{(m)}\}_{m=1}^M$ and $K$ test data sets $\{y^{(k)}\}_{k=1}^K$, we first compute the sampling distribution of MMDs from $M$ MMD estimates based on training samples $y$ vs. $y^{(m)}$. This quantifies the natural sampling distribution for $M$-vs.-1 MMD estimates where both samples stem from the training set. We then compute the $\alpha = 95\%$ percentile, which marks the cutoff for the $5\%$ most atypical training examples, and denote this threshold as $\text{MMD}_\alpha^2$. For the $K$ data sets in the test sample, we then compute the MMD estimate of all $M$ training samples against each of the $k = 1, \ldots, K$ test samples, here denoted as $\text{MMD}_k^2$. Then, we put it all together and flag data sets as atypical when $\text{MMD}_k^2 \geq \text{MMD}_\alpha^2$. The type-I error rate of this test can be set relatively high to obtain a conservative test that will flag many data sets for detailed investigation in further steps of our workflow.

**Note.** In the case study of this paper, we perform the above test in the summary space, that is, we replace all occurences of $y$ with the learned neural summary statistics $h_\psi(y)$, where $h_\psi$ is a DeepSet that learns an 8-dimensional representation of the data (see below for details).

# C Experiment details

In this section, we provide experiment details for parameter inference of the generalized extreme value (GEV) distribution.

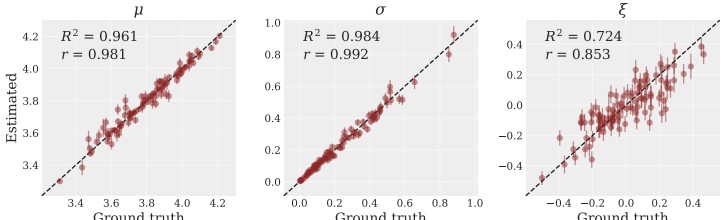

(a) The parameter recovery is excellent for the parameters $\mu, \sigma$ and good for the shape parameter $\xi$.



(b) Simulation-based calibration checking indicates excellent calibration for all parameters.

Figure 6: The closed-world diagnostics indicate acceptable convergence of the amortized posterior.

## C.1 Problem description

Following Caprani et al. [2], the prior distribution is defined as:

$$
\begin{aligned}
\mu &\sim \mathcal{N}(3.8, 0.04) \\
\sigma &\sim \text{Half-Normal}(0, 0.09) \\
\xi &\sim \text{Truncated-Normal}(0, 0.04) \text{ with bounds } [-0.6, 0.6].
\end{aligned}
\tag{6}
$$

## C.2 Simulation-based training

For the simulation-based training stage, we simulate $10\,000$ tuples of parameters and observations from the parameter priors and the corresponding GEV distributions. Each data set contains 65 *i.i.d.* observations from the GEV distribution. The validation set, generated in the same manner, consists of $1\,000$ samples from the joint model. The neural density estimator uses flow matching [13] as a generative neural network backbone. The internal network is a multilayer perception (MLP) with 5 layers of 128 units each, residual connections, and 5% dropout. Before entering the flow matching network as conditioning variables, we pre-process the observations $y = (y_1, \ldots, y_{65})$ with a DeepSet [25] that jointly learns an 8-dimensional embedding of the observations while accounting for the permutation-invariant structure of the data. The DeepSet has a depth of 1, uses a *mish* activation, max inner pooling layers, 64 units in the equivariant and invariant modules, and 5% dropout. In accordance with common practice in computational Bayesian statistics (e.g., PyMC or Stan), the amortized neural approximator learns to estimate the parameters in an *unconstrained* parameter space.

**Optimization.** The neural network is optimized via the Adam optimizer [12], with a cosine decay applied to the learning rate (initial learning rate of $10^{-4}$, a warmup target of $10^{-3}$, $\alpha = 10^{-3}$) as well as a global clipnorm of $1.0$. The batch size is set to $512$ and the number of training epochs is 300.

**Diagnostics.** The closed-world recovery (Figure 6a) and simulation-based calibration (Figure 6b) indicate that the neural network training has successfully converged to a trustworthy posterior approximator within the scope of the training set.

**Inference data sets** In order to emulate distribution shifts that arise in real-world applications while preserving the controlled experimental environment, we simulate the "observed" data sets from a joint model with a prior that has $4\times$ the dispersion of the model used during training. More

specifically, the prior is specified as:

$$\mu \sim \mathcal{N}(3.8, 0.16)$$
$$\sigma \sim \text{Half-Normal}(0, 0.36) \tag{7}$$
$$\xi \sim \text{Truncated-Normal}(0, 0.16) \text{ with bounds } [-1.2, 1.2].$$

### C.3 ChEES-HMC

We use $S = 16$ superchains and $L = 128$ subchains, resulting in a total number of $S \cdot L = 2048$ chains. The initial step size is set to 0.1. The number of warmup iterations is set to 200. The number of sampling iterations is 1, resulting in a total number of 2048 post-warmup MCMC draws.

