# OpenReview forum: "Amortized Bayesian Workflow (Extended Abstract)"
_NeurIPS.cc/2024/Workshop/BDU — NeurIPS BDU Workshop 2024 Poster_

### Official Review · Reviewer_4J8c · 2024-09-16
**The method description is clear, extending existing methods to DNNs, has some contributions, and includes experimental results.**

**Rating:** 6
**Confidence:** 3

**Review:**

Pros:
1.	The article is of good quality. It states that its contribution lies in the extension of the Bayesian workflow in deep learning. The author proposed a two-stage method based on amortized inference.

2.	The article is well presented and has a clear structure. The author can easily understand the content. There are figures in the intro and method sections that help readers understand. In each of the two stages, the paper illustrates the method in several steps.

Cons:
1.	About contribution: It mainly integrates amortized inference into the Bayesian workflow and applies it to deep learning. However, the training stage is similar to the common method, the inference stage mainly incorporates the amortized concept.
2.	My concern is mainly in the training stage. The author points out that a sufficient training budget is needed to ensure the effect of the amortized posterior estimator. Then the cost of this method in the actual application of DNNs is relatively large, although it is fast on the existing experiments.
3.	I hope that the author can continue to improve the method and apply it to more specific deep learning settings. It is helpful to show that the proposed method has enough significance.

---

### Official Review · Reviewer_5om8 · 2024-09-22
**Review for Paper "Amortized Bayesian Workflow"**

**Rating:** 7
**Confidence:** 3

**Review:**

The paper presents an adaptive approach that integrates amortized Bayesian inference into traditional Bayesian workflows. It introduces an adaptive workflow that produces high-quality posterior samples while minimizing computational time. The empirical demonstration of the method focuses on a generalized extreme value distribution, highlighting efficiency improvements without compromising inference quality.

The work aligns well with the theme of the workshop, and the manuscript is structured in a logical and well-organized manner.

Overall, I recommend accept to the workshop, and it is encouraged to extend the paper with more theoretical analysis and empirical evaluations.

---

### Decision · Program_Chairs · 2024-10-09

Accept (Poster)